# Microwave excitation of atomic scale superconducting bound states

Janis Siebrecht[1], Haonan Huang [1], Piotr Kot[1], Robert Drost[1], Ciprian Padurariu [2], Björn Kubala [2,3], Joachim Ankerhold [2], Juan Carlos Cuevas [4] & Christian R. Ast [1] ✉

Magnetic impurities on superconductors lead to bound states within the superconducting gap, so called Yu-Shiba-Rusinov (YSR) states. They are parity protected, which enhances their lifetime, but makes it more difficult to excite them. Here, we realize the excitation of YSR states by microwaves facilitated by the tunnel coupling to another superconducting electrode in a scanning tunneling microscope (STM). We identify the excitation process through a family of anomalous microwave-assisted tunneling peaks originating from a second-order resonant Andreev process, in which the microwave excites the YSR state triggering a tunneling event transferring a total of two charges. We vary the amplitude and the frequency of the microwave to identify the energy threshold and the evolution of this excitation process. Our work sets an experimental basis and proof-of-principle for the manipulation of YSR states using microwaves with an outlook towards YSR qubits.

Magnetic impurities coupled to a superconductor give rise to Yu-Shiba-Rusinov (YSR) states, which are subgap states protected by parity (even/odd particle number conservation)[1–3]. They exhibit a variety of interesting phenomena including (but not limited to) their resonant character, which enhances higher order processes in tunneling (Andreev processes) or their parity protection, which enhances their lifetime[4–6]. Comparatively long coherence times can also be expected in YSR states, but work on a coherent coupling of YSR states so far has been limited[4,7]. The first step towards coherent manipulation is the use of microwaves in a tunnel junction, which leads to microwave-assisted tunneling[8–10]. However, parity conservation has to be considered when exciting a YSR state using microwaves.

Elementary excitations in a superconductor, i.e. Bogoliubov quasiparticles, come in pairs due to parity conservation, but only one quasiparticle is needed to excite the YSR state[11]. The second quasiparticle can escape to the continuum, which requires excitation energies of at least the superconducting gap, or through a tunneling contact, where much lower excitation energies are sufficient. A scanning tunneling microscope (STM) provides such a tunneling contact offering the ability to manipulate a YSR state with moderate excitation energies far below the superconducting gap. This makes the STM an ideal platform for the manipulation of YSR states as an extension of nondegenerate Andreev-bound states to the atomic scale[12], which provides a starting point for YSR qubits[13–17].

Here, we demonstrate the excitation of YSR states using microwaves in the tunnel junction of an STM. We are able to separate different tunneling processes involving the YSR states, which allows us to identify a tunneling process that is only possible through the direct excitation of a YSR state by the microwave. We map out an amplitude threshold that has to be overcome to excite the YSR state. This threshold depends on the applied bias voltage, which allows for great flexibility in different YSR excitation schemes. In this way, we provide a proof of principle for the excitation and manipulation of YSR states by microwaves in the presence of a tunnel junction, which is an important prerequisite for the preparation and control of complex YSR structures, for example, in the context of quantum simulations.

We use a scanning tunneling microscope with an external microwave antenna optimized for operation between 60 GHz and 90 GHz[18],

[1]Max-Planck-Institut für Festkörperforschung, Heisenbergstraße 1, 70569 Stuttgart, Germany. [2]Institut für Komplexe Quantensysteme and IQST, Universität Ulm, Albert-Einstein-Allee 11, 89069 Ulm, Germany. [3]Institute for Quantum Technologies, German Aerospace Center (DLR), Wilhelm-Runge Straße 10, 89081 Ulm, Germany. [4]Departamento de Física Teórica de la Materia Condensada and Condensed Matter Physics Center (IFIMAC), Universidad Autónoma de Madrid, 28049 Madrid, Spain. ✉e-mail: c.ast@fkf.mpg.de

which is schematically shown in Fig. 1a. By controlled dipping of a vanadium tip in a V(100) surface, we create a YSR state at the apex of the tip[4,19], which is subsequently irradiated by microwaves (see Supplementary Note 1). In Fig. 1b, the differential conductance (green line) through a YSR state in the absence of microwaves is shown. The salient features of the YSR state are two sharp peaks in the superconducting gap at $eV = \pm(\Delta_s + \varepsilon)$ corresponding to the electron and hole parts of the Bogoliubov quasiparticle ($V$ is the bias voltage, $\Delta_{t,s}$ is the superconducting gap parameter in tip and sample, and $\varepsilon$ is the YSR energy).

In recent experiments, microwaves have successfully been implemented in STMs with various applications, such as resolving the internal structure of complex tunneling processes. Initial experiments on clean superconductors[10] show good agreement with a theory for microwave-assisted tunneling[20,21], which we refer to in the following as Tien-Gordon (TG) theory. This theory predicts the formation of replicas of very sharp spectral features (e.g. coherence peaks, YSR states) at integer multiples of $\hbar\omega_r/e$ weighted by a squared Bessel function ($\omega_r$ is the microwave radiation frequency, $\hbar$ is Planck's constant, and $e$ is the elementary charge), which depends on the microwave amplitude. Further work has shown that this theory needs to be generalized beyond the tunneling regime for higher order processes such as the Josephson effect or Andreev reflections[8]. For a non-resonant transfer of $n$ charges, replicas form at multiples of $\hbar\omega_r/ne$[22-24]. Also, it has been demonstrated that replicas of YSR states can show asymmetries which are not contained within the TG theory[9]. This was corroborated by a simplified Green's functions approach[25].

## Results and discussion
### Exciting YSR States with microwaves
The microwaves induce an alternating voltage $V_{ac}$ in the tunnel junction, which is on the order of 100 $\mu V$ to 10 mV. The conductance spectrum with a YSR state irradiated by microwaves at a frequency of $\omega_r/2\pi = 60.05$ GHz and an amplitude of 570 $\mu V$ is shown in Fig. 1c (yellow green line). We note that the temperature of the thermometer at the STM only increases by a few mK, which leads us to conclude that

the superconducting junction is unaffected in line with previous work[8]. Furthermore, in the modeling we achieved excellent agreement by assuming a constant temperature of 560 mK independent of the microwave amplitude. The interaction of the tunneling electrons with the microwave leads to both the absorption and emission of energy quanta by the tunneling electrons in integer multiples of $\hbar\omega_r = 248.3 \mu eV$. In the simplest approximation, this interaction leads to the appearance of replicas of the spectral features in Fig. 1b. In Fig. 1c, the expected replicas of the YSR states are indicated by blue vertical lines at distances of 248.3 $\mu V$. However, we also observe a number of additional peaks marked by the red vertical lines, which appear at $eV = \pm(\Delta_s - \varepsilon) + n\hbar\omega_r$, where $n$ is an integer. This might suggest a thermal origin, but the temperature of 560 mK is very low and no corresponding peak can be seen in the spectrum in the absence of microwaves (cf. Fig. 1b).

### Ground state vs. excited state tunneling
To understand the origin of the different peaks seen in Fig. 1c, we present schematics of the underlying tunneling processes in Fig. 1d–g. To induce tunneling through the YSR state without microwaves, we apply a bias voltage of $eV = \Delta_s + \varepsilon$ as shown in Fig. 1d. To illustrate this, we divide the tunneling process into three steps using the density of states picture. In the first step (labeled ①), an electron is transferred across the tunnel junction. In the second step (labeled ②) a Cooper pair is split filling the hole, but leaving the YSR state excited. This excited quasiparticle then relaxes into the continuum (step ③') or tunnels across the junction as well (step ③). If the tunnel coupling is weak, quasiparticle relaxation in the YSR electrode dominates (step ③'). As the tunnel coupling increases, step ③ becomes dominant transferring a total of two charges across the junction. This step is termed "resonant Andreev process" as its tunneling path involves a real state (the YSR state[6,9,26]) instead of a virtual state as in conventional Andreev reflections[27]. We note that higher-order transfer processes appear in resonant tunneling processes at much lower conductances than for "conventional" tunneling, e.g. Andreev reflections. Therefore, a

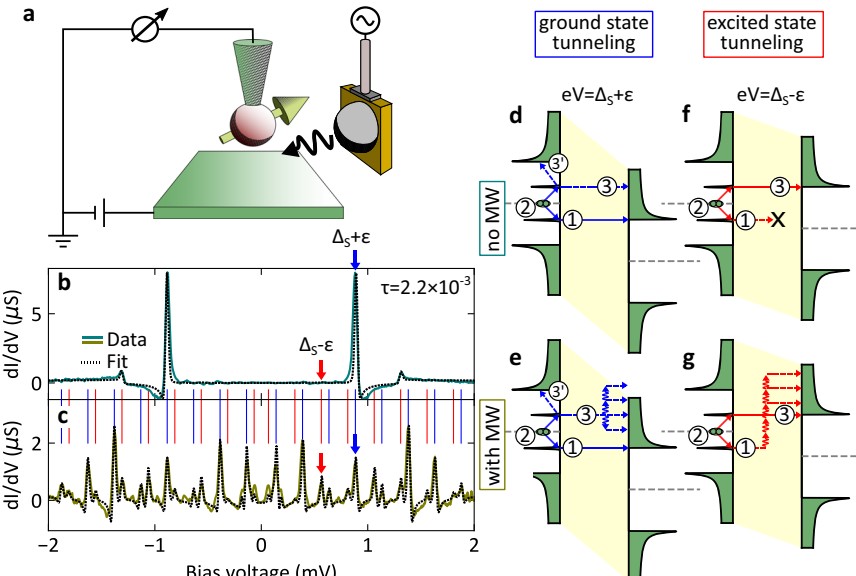

**Fig. 1 | Tunneling mechanisms of YSR states under microwave irradiation.**
**a** Schematic drawing of the experimental setup. **b** Differential conductance measured without microwaves. Ground state tunneling is indicated by a blue arrow. No excited state tunneling is observed (red arrow). **c** Differential conductance measured with microwaves at 61 GHz. The energy exchange with the microwave induces replicas. The zero order ground state tunneling is indicated by a blue arrow. Excited state tunneling induces additional peaks, with the zero order peak indicated by a red arrow. The dashed lines in (**b**) and (**c**) are fits to the data using the full Green's function model and two transport channels (one BCS and one YSR channel (cf.[4,19])). **d–g** Schematics illustrating ground state and excited state tunneling processes with and without microwaves. The schematics are drawn for the zero-order processes, i.e. no net energy quanta transferred. Energy quanta may be absorbed/emitted in steps ①/③ leading to replicas at different bias voltages.

theoretical description has to include these processes already at a conductance of $2.2 \times 10^{-3} G_0$, where Andreev reflections can still be neglected ($G_0 = 2e^2/h$ is the quantum of conductance states). The event illustrated in Fig. 1d leads to a spectral peak indicated by the blue arrow in Fig. 1b.

In the presence of microwaves, the tunneling process indicated by the blue arrow in the experimental spectrum in Fig. 1c is schematically shown in Fig. 1e. We first observe the conventional peak to appear at a bias voltage of $eV = \Delta_s + \varepsilon$, which implies that a total of zero energy quanta are exchanged with the microwave during step ①. However, energy quanta can be exchanged during step ③, yet without shifting the position of the peak.

In fact, the peak position only changes if energy quanta are absorbed or emitted during step ① such that they appear at different bias voltages $eV = \pm (\Delta_s + \varepsilon) + n\hbar\omega_r$ in the spectrum. Other than that, the process is analogous to the tunneling without microwaves (cf. Fig. 1d). In the following, we consider the two processes involving step ③ and ⑤ together and refer to this family of peaks as ground state tunneling.

The additional peaks seen as red lines in Fig. 1c cannot be explained by ground state tunneling (cf. Fig. 1d, e). They can be attributed to processes which originate from tunneling events in the absence of microwaves at bias voltages of $eV = \Delta_s - \varepsilon$ as depicted in Fig. 1f, where we would expect them to occur via thermal activation. However, in our experiment, the Boltzmann factor $\exp(-\frac{\varepsilon}{k_B T})$ for a YSR state of energy $\varepsilon = 280 \ \mu V$ (for Figs. 2–4) at a temperature of 0.56 K predicts a contribution of 0.03%, so that thermal excitations are strongly suppressed. Indeed, Fig. 1b shows no spectral feature, where the red arrow is pointing. When we turn on the microwaves, a strong and clear peak can be observed at the location of the red arrow, in contrast to a strong peak in the presence of microwaves in Fig. 1c. In this situation, the microwaves open new transfer channels as delineated in Fig. 1g. The absorption of multiple energy quanta during step ① induces an excited YSR state (step ②) and allows for subsequent relaxation into the continuum through step ③. Multiple quanta being absorbed or emitted during process ③ then lead to a family of additional peaks marked by the red lines in Fig. 1c at bias voltages $eV = \pm (\Delta_s - \varepsilon) + n\hbar\omega_r$. All the peaks of this family have in common that the excited state is aligned with the coherence peak through the bias voltage modulo an integer number of microwave quanta, which is why we call these processes excited state tunneling.

## Modeling the microwave excitation of YSR states

In order to understand the evolution of the ground state and excited state tunneling more quantitatively, we measure differential conductance spectra as a function of the dimensionless microwave amplitude $\alpha = eV_{ac}/\hbar\omega_r$. Figure 2a shows the differential conductance measured at a microwave frequency of 61 GHz and a normal state conductance of $G_N = 2.2 \times 10^{-3} G_0$, where $G_0 = 2e^2/h$ is the quantum of conductance. We can clearly see many well-defined peaks, which we will assign to ground state or excited state tunneling in the following. In order to distinguish these peaks, we use the TG model to calculate the expected microwave amplitude dependence from the measured conductance spectrum without microwaves[10,20]

$$I(V, \alpha) = \sum_n J_n^2(\alpha) I^0 \left(V + n\hbar\omega_r/e\right), \tag{1}$$

where $J_n(\alpha)$ is the $n$th order Bessel function of the first kind and $I^0(V)$ is the tunneling current without microwaves. The calculated image starting from the zero amplitude spectrum in Fig. 2a is shown in Fig. 2b. We note that the TG model does not reproduce all of the experimentally observed peaks. The replicated peaks in Fig. 2b are entirely due to ground state tunneling, so that all additional peaks in Fig. 2a must be due to excited state tunneling. For comparison, we calculate the data set in Fig. 2a using the full Green's function theory taking into account microwaves, higher order tunneling processes (e.g. Andreev processes) as well as the interference between them[23] (for details see Supplementary Note 2). We found that due to the resonant tunneling through the YSR states the interplay between the microwave and the higher order tunneling processes become non-negligible such that approximative calculations fail and the full Green's function model has to be applied (for details see Supplementary Note 3). The calculation is shown in Fig. 2c, which shows excellent agreement with the measured data in Fig. 2a. Both ground state and excited state tunneling processes are reproduced with the full Green's function model.

## Frequency dependence of YSR excitations

To substantiate our claim that there are indeed two families of processes, we present frequency-dependent differential conductance spectra at a constant dimensionless amplitude of $\alpha = 3$ in Fig. 3a. The higher the order of the replica, the more tilted the spectral feature will appear in the map. An $n$th order replica of a feature at $V_0$ moves as $eV = eV_0 + n\hbar\omega_r$. The replica and their dispersion are calculated from the full Green's function theory in Fig. 3b as well as presented schematically in Fig. 3c. We can identify four vertical lines corresponding to zero order replicas, marked by the lines connecting panels (a), (b), and (c). The blue and red colors mark ground state tunneling ($eV_0 = \pm (\Delta_s + \varepsilon)$) and excited state tunneling ($eV_0 = \pm (\Delta_s - \varepsilon)$), respectively. We note that at $\alpha = 3$, the microwave has enough power to excite the YSR state, such that excited state tunneling becomes possible. If the excited state replica actually did appear in the spectrum without microwaves, which is not the case (cf. Fig. 1b), the original spectrum would appear as in Fig. 3d. In Fig. 3d, the excited state tunneling peak (red line) is added manually, where thermal

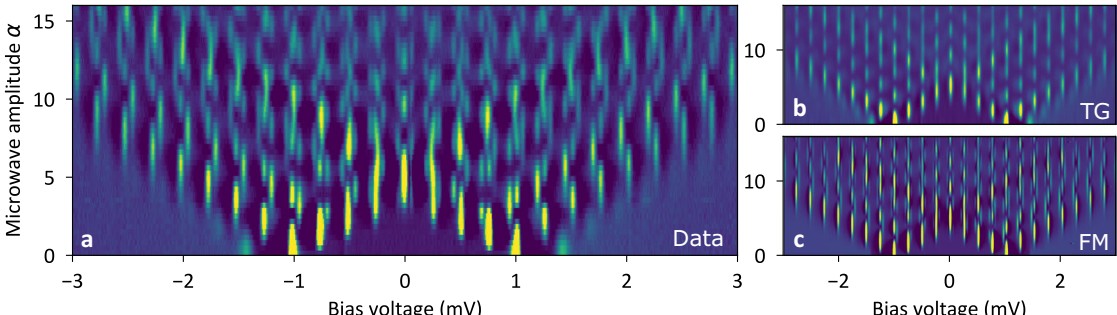

**Fig. 2 | Differential Conductance as function of bias of bias voltage and microwave amplitude. a** Experimental data measured at a setpoint of 500 pA at 3 mV with a microwave frequency 61 GHz. **b** Calculation based on the spectrum at zero amplitude in (**a**) using the Tien–Gordon (TG) model. The features connected to excited state tunneling are missing. **c** Full Green's function model (FM) calculation shows all details as in the experimental data.

tunneling would appear, as a Gaussian peak with the same width and height as the corresponding YSR peak. However, microwaves could trigger these transfer processes to occur beyond a given threshold as discussed below.

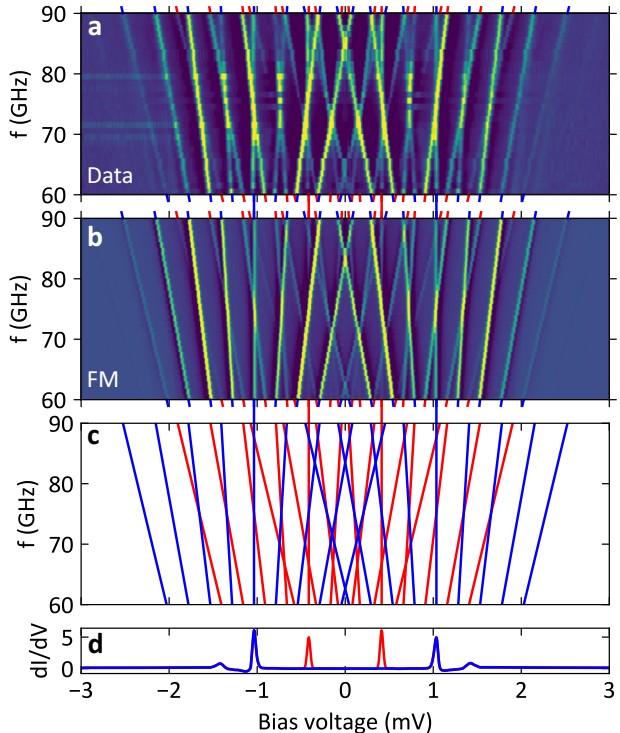

**Fig. 3 | Frequency dependence of the spectra at constant microwave amplitude α. a** Differential conductance spectra measured as a function of frequency at constant microwave amplitude $\alpha = \frac{eV_{ac}}{\hbar\omega_r} = 3$. **b** Calculated spectra in the same range as (**a**) (full model). **c** Theoretical location of normal state replicas (blue) and excited states replicas (red). **d** Base spectrum without microwaves (blue) and excited states represented by a Gaussian peak, which was added manually (red), having the same height and width as the corresponding YSR peak. The zeroth order replicas (vertical lines) connect the panels.

## A simple model for excited state tunneling

In essence, the breakdown of the simple TG model (Fig. 2b) is expected because it leaves the ground state untouched and only considers the spectrum in the absence of microwaves without taking into account processes activated by the microwaves, such as excited state tunneling. The full model (Fig. 2c) agrees quantitatively with the experiment (cf. excellent fit in Fig. 1c). An intuitive understanding of the mechanism behind excited state tunneling can be derived from a simplified model. Employing a perturbative approach including second-order resonant Andreev processes, the excited state tunneling current $I_{ex,e/h}(V, \alpha)$ appears as

$$I_{\mathrm{ex,e/h}}(V, \alpha) = \sum_n w(\alpha, n) J_n^2(\alpha) I^0_{\mathrm{ex,e/h}}(V \pm n\hbar\omega_r/e), \quad (2)$$

where e/h refers to the peak at negative/positive bias voltage $eV = \mp(\Delta_s - \varepsilon)$. The bare excited state tunneling current $I^0_{ex,e/h}(V)$ is replicated by the microwave beyond an amplitude threshold (see below). This is detailed in Supplementary Note 4 along with details on the approximations being used. We further introduce a weight function

$$w(\alpha, n) = \sum_{m \geq m_0 - n} J_m^2(\alpha), \quad (3)$$

which sums over all possible energy quanta that can be exchanged during step ①, where $m_0 = \left\lceil \frac{2\varepsilon}{\hbar\omega_r} \right\rceil$ is the minimum number of quanta needed to excite the YSR state (cf. Fig. 1g). The excited state tunneling current in Eq. (2) and the weight function in Eq. (3) show that step ① in Fig. 1g only contributes to the magnitude of the current, but it does not generate any replica. This also explains why the replica are a distance $\hbar\omega_r/e$ apart despite two charges being transferred in the whole process. A very similar argument can be made for step ③ of the ground state tunneling in Fig. 1e. However, in this case the sum condition in the weight function is $m > n - m_0$, which does not introduce a new threshold, but just leads to a renormalization of the spectral weight. A number of different approximations between the full model and the simplified model in Eq. (2) can be made, e.g.[25], which are discussed in Supplementary Notes 2, 3, and 4.

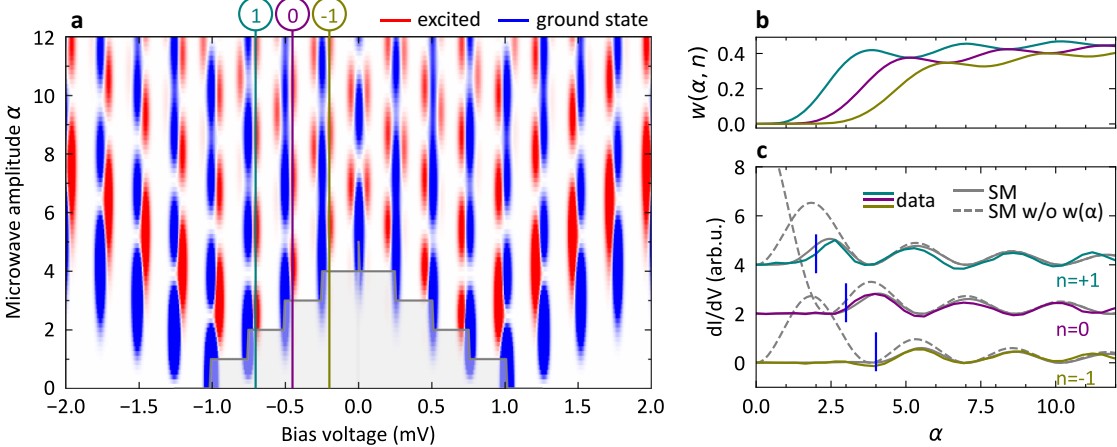

**Fig. 4 | Excitation threshold for the YSR state. a** Differential conductance calculation (simplified model) to illustrate the origin of the tunneling processes. Ground state tunneling is shown in blue and excited state tunneling in red. The shaded area around zero bias voltage represents the region where the microwave amplitude is below the threshold to activate excited state tunneling. **b** The weight function for excited state tunneling for different orders $n = -1, 0, 1$ as indicated by the vertical lines in (**a**). The initial threshold is clearly visible. **c** Slices of differential conductance data as a function of microwave amplitude for excited state tunneling at different orders $n = -1, 0, 1$ as indicated by the vertical lines in (**a**). The simplified model (SM) (solid gray line) fits well with the experimental data and nicely demonstrates the cutoff due to the threshold at lower amplitudes compared to when the weight function is not considered (gray dashed line). The vertical blue lines indicate the threshold $\alpha = 2, 3, 4$, respectively.

In Fig. 4a, we separately calculate the ground state and excited state tunneling conductances using Eq. (2) and the corresponding formula for ground state tunneling (see Supplementary Note 4), which are shown in blue and red, respectively. The stepped shaded area around zero bias voltage represents the threshold amplitude needed to activate excited state tunneling. The weight function for the $n = -1, 0, 1$ processes marked in Fig. 4a are plotted in Fig. 4b as a function of dimensionless microwave amplitude $\alpha$. We see that the threshold is not a sharp cutoff, but follows the leading edge of the lowest order Bessel function $J^2_{2,3,4}(\alpha)$, respectively, enabling the process. For $n = -1, 0, 1$, the threshold is roughly at $\alpha \gtrsim 2, 3, 4$, respectively, when the weight function becomes significant. To demonstrate the threshold effect of the weight function, we plot the corresponding data from Fig. 2a at $n = -1, 0, 1$ in Fig. 4c. The fits are shown with and without the weight function as solid and dashed gray line, respectively. We can directly see how the weight function imposes the threshold for small amplitudes and nicely follows the experimental data.

At this point, we emphasize that here YSR states are excited using energies much smaller than the minimal energy $\Delta E > \Delta_s + \varepsilon$, if the YSR state is connected to a tunnel junction. In fact, the excitation energy can be as low as $2\varepsilon$ (cf. Fig. 1g and Eq. (2)), which we have demonstrated through the excited state tunneling process and the imposed activation threshold. Even though two electrons are transferred in the resonant tunneling process through the YSR state, the replica are spaced $\hbar\omega_r/e$ apart instead of $\hbar\omega_r/2e$ as for conventional Andreev reflections. Hence, the spacing between replica cannot be used for inferring the number of charges being transferred. Our ability to excite YSR states with high precision can now be exploited for direct manipulation protocols. This opens up new possibilities for pump-probe schemes to address the finite lifetime of YSR states.

In summary, we have conducted a proof-of-principle experiment showing that the combination of a tunneling current and microwave radiation can excite YSR states without the need to cross the energy gap. In particular, microwave-assisted tunneling can be used as a tool not only for ground-state tunneling, but also for excited-state tunneling. The sub-gap excitation is attractive for future applications (such as information storage) as it does not introduce decoherence by coupling to the continuum to which the YSR state is coupled. Therefore, microwaves could pave the path towards coherent manipulation, similar to ESR-STM[28] or Andreev qubit architectures[14]. Additionally, this work has shown replicas at multiples of $\hbar\omega_r/e$ as opposed to the $\hbar\omega_r/2e$ that one would expect for a two-electron process. Pulse schemes or shot noise measurements[29,30] could shed further light on this process.

## Data availability
The data that support the findings of this study are available from the corresponding author upon request.

## Code availability
The computer code that support the findings of this study is available from the corresponding author upon request.

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

## Acknowledgements
The authors thank Klaus Kern for fruitful discussions. This study was funded in part by the ERC Consolidator Grant AbsoluteSpin (Grant No. 681164). J.C.C. thanks the Spanish Ministry of Science and Innovation (Grant No. PID2020-114880GB-I00) for financial support as well as the DFG and SFB 1432 for sponsoring his stay at the University of Konstanz as a Mercator Fellow.

## Author contributions
J.S. measured the data with support from H.H., P.K., and R.D. J.C.C., C.P., B.K., and J.A. provided theory support. J.S. and C.R.A. analyzed the data with support from all authors. J.S. and C.R.A. wrote the manuscript with input from all authors.

## Funding

## Competing interests
The authors declare no competing interests.
