## [Peer Review File · Nature Communications]

Microwave Excitation of Atomic Scale Superconducting Bound StatesREVIEWER COMMENTS

Reviewer #1 (Remarks to the Author):

The manuscript by Janis Siebrecht et al. is a beautiful work that reports on the investigation of tunnel spectroscopy data of a single atomic spin attached to the superconducting tip of a scanning tunnel microscope against a superconducting sample which is irradiated by microwaves. It reports the first and very compelling evidence of a controlled excitation of the Yu-Shiba-Rusinov (YSR) states bound to the atomic spin coupled to a superconductor by means of microwaves. The main experimental observation is the emergence of a new family of equidistant peaks in the tunnel spectra once the system is irradiated by microwaves. Via a very systematic investigation of the dependence of the energies and intensities of these peaks in the tunnel spectra (taken at sub-Kelvin temperatures!) as a function of microwave amplitude and frequency, the authors achieve a beautiful data set which can be directly compared to three different models. The simulation with a Full Green's function model shows an excellent agreement with the experimental data unambiguously leading to the conclusion that the additional peaks can only be explained by the microwave excitation of the YSR states in the tunnel gap via a second order resonant Andreev process.

The main result of the manuscript, i.e. the successful microwave-excitation of YSR states of atomic spins coupled to superconductors, opens up possibilities for a vast bunch of experiments related to many ideas on the coherent manipulation of spins coupled to superconductors towards the goal of the realization of a YSR qubit. The work is not only interesting for the scientific community working on atomic spins on surfaces, but also to researchers working in the broader area of qubits in contact to superconductors, as e.g. semiconductor quantum dots with superconducting leads. I am, therefore, convinced that the work is of sufficiently high significance, novelty and interest to warrant publication in a Journal with a broad readership like Nature Communications. Overall the analysis of the experimental data is technically sound and the scholar presentation of the results is of exceptional quality. I have only a few minor comments/criticisms which I will explain below point-by-point, and recommend the publication of the manuscript after a consideration of these minor points and a final revision.

- 1) In the second paragraph of the introduction, the authors relate their work to YSR qubits. They should consider adding a reference to the following work, which seems to be relevant in this respect: PRX Quantum 2, 040347 (2021).
- 2) There are some additional weak peaks in the tunnel spectrum in Fig.1(c) which are not reproduced by the full Green's function model and do not coincide neither with the red nor with the blue vertical lines, e.g. at about -0.7mV, -1mV, and -1.2mV. Is there an understanding, where these peaks come from? It would be beneficial for the reader to mention these (though slight and probably negligible) deviations from the model.
- 3) In the caption of Fig.1 it is stated that microwave energy quanta may be absorbed/emitted in steps 1 and 3 of (d) to (g). However, for step 1, these processes are only considered for the case of Fig.1(g) and not for the case of Fig.1(e). Why are they missing in the latter case. Shouldn't these processes be also present for that case and even are essential in order to explain the family of peaks marked with the blue vertical lines in Fig.1(c)? This is also related to the text passages in the second paragraph of page 3, left column, and to the following sentence on page 5, left column: "The excited state tunneling current in Eq. (2) and the weight function in Eq. (3) show that step 1 in Fig. 1(g) only contributes to the magnitude of the current, but it does not generate any replica".
- 4) Page 2, right column, bottom line: delete the repeated word "processes".
- 5) In the caption of Fig.3, the "manual addition of the excited states" in Fig.3(d) is an unclear description of where these spectra have been taken from. Please describe more clearly.
- 6) Page 3, right column, 5th line: add "we" between "why" and "call".
- 7) There are gray shaded areas in Fig.4(a), but the meaning and origin of these areas is not described in the caption. Maybe I overlooked it. They probably indicate the threshold values for α taken from Fig.4(c)?
- 8) Supplementary Information, Section IV.: Figs.S4 and S5 should be already referred to in the end of the first paragraph, as the text in the following enumeration of the three different models

refers to black, blue, and red lines, respectively.

9) Supplementary Information, Equation (S39): The superfluous "d" in the integral in the denominator should be deleted.

Reviewer #2 (Remarks to the Author):

The manuscript reports on the observation of resonant Andreev tunneling processes between a Shiba state located on the superconducting tip and a superconducting substrate.

Such resonant processes have been observed previously by the same group (Huan et al., Karan et al, Kot et al) and other (Peters et al; Ruby et al.).

This resonant tunneling process can occur at an energy $\Delta + \varepsilon$, leaving the Shiba into its ground state or occurs at $\Delta - \varepsilon$, leaving the Shiba into its excited state.

Both processes have been observed previously, in particular (Peters et al; Ruby et al.). In those past experiments, the excited state could only be reached through thermal excitations.

In the present manuscript, the authors show that the excited state can be reached at very low temperature by applying a microwave signal and concluded that this observation is interesting as it paves the way toward using microwave signals to manipulate the quantum state of the Shiba.

The work is of high quality, well written and consistent with past works of the same group and others. While the manuscript presents only a small additional novelty with respect to previous works, I recommend publication but I suggest that the authors complete the manuscript along these lines :

- it would be useful to add details about the nature of the magnetic impurity in the present manuscript, to avoid going back to past published works.
- with respect to quantum manipulation, the orbital and spin degeneracy of the Shiba is important. In Hay et al. works on nanowires, spin-orbit coupling is needed to lift the spin-degeneracy. What is known about the orbital and spin degeneracies of the Shiba state in the present manuscript.
- Providing a sketch of the protocol that could be employed to produce quantum manipulation of the Shiba state to observe, for example, Rabi oscillations, would reinforce the claim that quantum coherent manipulation of the Shiba state with microwave signals is possible.

minor corrections are :

Figure 1 : "Tunneling" written with three n

« proof-of-principle » i missing

Reviewer #3 (Remarks to the Author):

The authors of this collaboration report excitation of Yu-Shiba-Rusinov states in an atomic-scale tunneling junction using microwaves. Along with careful measurements of the excitation spectra, they provide a thorough theoretical analysis. This analysis goes beyond the previously used Tien-Gordon theory and accounts for most features in the spectra and, in particular, the series of (red) peaks identified in Fig. 1b. The latter are explained in terms of tunneling mediated by excited states.

Although this experiment is not the first of its kind and similar results have been reported in Ref. 9 by Peters and coworkers, the quality of the experimental data is impressive and the theoretical analysis thereof is fairly thorough (within the assumptions made by the authors, see below). Before publication may be granted, the authors should provide further explanations on a number of issues described below.

The main issue with this study concerns the rather "ad hoc" preparation of the "magnetic impurity" in the tip of the STM, which is subsequently studied under microwave irradiation. The preparation procedure is briefly described in the first section of the supplementary information (SI), but additional details are necessary. This group has used this method in previous work and they refer to this fact in the manuscript. However, in order to fully justify their theoretical treatment _independently of the fact that the latter agrees well with the experimental observations_ a reassessment of some of their assumptions would be most welcome. For example, what is the spin of the impurity? In their theoretical model, the impurity is described using a minimal Anderson model in the mean-field approximation (Refs. 1 and 6 of the SI), which means the impurity is treated as a classical local exchange potential. However, in a fully quantum mechanical treatment, the impurity orbital may contain more than one electron resulting in higher spin and other terms like single-ion anisotropy. This would lead to additional excitation channels in the presence of microwaves. What is the experimental evidence that such channels are not present or relevant? More specifically, what would be the difference between a spin-1/2 impurity as assumed and a higher spin impurity with positive single-ion anisotropy for which only its lowest two spin states are available in the absence of microwaves but higher spin states being excited upon irradiation? Can the authors entirely rule out that the internal excitations of the impurity may account for the experimental observations?

The authors also provide some evidence that supports their interpretation that processes responsible for the red peaks involve excited states. However, the discussion of why it is possible to neglect thermally activated processes does not appear entirely convincing. The argument found on page 3 seems to rely on the assumption that the temperatures of irradiated and non-irradiated junctions stays roughly the same (0.56K). However, irradiation is expected to cause heating of the junction, as the authors themselves acknowledge on page 2. Therefore, they should provide an improved discussion of this point beyond the sentence (page 2) "We note that the temperature of the junction only increases by a few mK, which we can safely assume to be constant in line with previous work [8]". Can they possibly explain, for instance, how this "few mK" estimate obtained? And why should the distribution of the irradiated atomic junction remain thermal at all?

Typos: in the last paragraph of page 2, the word "processes" appears twice.

Reply to Referee's Comments

Changes in the manuscript are marked in blue.

Answers to Referee #1:

The manuscript by Janis Siebrecht et al. is a beautiful work that reports on the investigation of tunnel spectroscopy data of a single atomic spin attached to the superconducting tip of a scanning tunnel microscope against a superconducting sample which is irradiated by microwaves. It reports the first and very compelling evidence of a controlled excitation of the Yu-Shiba-Rusinov (YSR) states bound to the atomic spin coupled to a superconductor by means of microwaves. The main experimental observation is the emergence of a new family of equidistant peaks in the tunnel spectra once the system is irradiated by microwaves. Via a very systematic investigation of the dependence of the energies and intensities of these peaks in the tunnel spectra (taken at sub-Kelvin temperatures!) as a function of microwave amplitude and frequency, the authors achieve a beautiful data set which can be directly compared to three different models. The simulation with a Full Green's function model shows an excellent agreement with the experimental data unambiguously leading to the conclusion that the additional peaks can only be explained by the microwave excitation of the YSR states in the tunnel gap via a second order resonant Andreev process.

The main result of the manuscript, i.e. the successful microwave-excitation of YSR states of atomic spins coupled to superconductors, opens up possibilities for a vast bunch of experiments related to many ideas on the coherent manipulation of spins coupled to superconductors towards the goal of the realization of a YSR qubit. The work is not only interesting for the scientific community working on atomic spins on surfaces, but also to researchers working in the broader area of qubits in contact to superconductors, as e.g. semiconductor quantum dots with superconducting leads. I am, therefore, convinced that the work is of sufficiently high significance, novelty and interest to warrant publication in a Journal with a broad readership like Nature Communications. Overall the analysis of the experimental data is technically sound and the scholar presentation of the results is of exceptional quality. I have only a few minor comments/criticisms which I will explain below point-by-point, and recommend the publication of the manuscript after a consideration of these minor points and a final revision.

We thank the reviewer for their positive assessment. We are happy to hear that they share our conviction of the implications of this work.

1) In the second paragraph of the introduction, the authors relate their work to YSR qubits. They should consider adding a reference to the following work, which seems to be relevant in this respect: PRX Quantum 2, 040347 (2021).

We thank the reviewer for pointing us to this reference, which we have included now.

2) There are some additional weak peaks in the tunnel spectrum in Fig.1(c) which are not reproduced by the full Green's function model and do not coincide neither with the red nor with the blue vertical lines, e.g. at about -0.7mV, -1mV, and -1.2mV. Is there an understanding, where these peaks come from? It would be beneficial for the reader to mention these (though slight and probably negligible) deviations from the model.

Indeed, these peaks are replica of the small coherence peak at Δ , which appears due to a finite filling of the gap (elevated Dynes parameter). This is not due to Andreev reflections, for which the junction transmissio is to low. We did not model them in detail because this part would not change the interpretation of the results. We have added a short statement in the supporting information to explain this.

3) In the caption of Fig.1 it is stated that microwave energy quanta may be absorbed/emitted in steps 1 and 3 of (d) to (g). However, for step 1, these processes are only considered for the case of Fig.1(g) and not for the case of Fig.1(e). Why are they missing in the latter case. Shouldn't these processes be also present for that case and even are essential in order to explain the family of peaks marked with the blue vertical lines in Fig.1(c)? This is also related to the text passages in the second paragraph of page 3, left column, and to the following sentence on page 5, left column: "The excited state tunneling current in Eq. (2) and the weight function in Eq. (3) show that step 1 in Fig. 1(g) only contributes to the magnitude of the current, but it does not generate any replica".

This observation is in principle correct: during step 1 absorption of other quanta is also possible. However, at the given bias voltage $eV = \Delta_S + \epsilon$, as indicated at the top of the figure, these additional transitions would not really contribute to the peak observed in the dI/dV -spectrum. So, for simplicity we are only considering the process which are in resonance at $eV = \Delta_S + \epsilon$. A similar diagram can indeed be drawn for $eV = \Delta_S + \epsilon + n\hbar\omega_r$, where n quanta are absorbed during process 1, which would correspond to the other blue vertical lines in Fig. 1c.

4) Page 2, right column, bottom line: delete the repeated word "processes".

Done.

5) In the caption of Fig.3, the "manual addition of the excited states" in Fig.3(d) is an unclear description of where these spectra have been taken from. Please describe more clearly.

We have added a short sentence that we manually add Gaussian peaks with corresponding height and width.

6) Page 3, right column, 5th line: add "we" between "why" and "call".

Done.

7) There are gray shaded areas in Fig.4(a), but the meaning and origin of these areas is not described in the caption. Maybe I overlooked it. They probably indicate the threshold values for α taken from Fig.4(c)?

This is correct and already described in the main text. "The stepped shaded area around zero bias voltage represents the threshold amplitude needed to activate excited state tunneling." We have added a statement in the figure caption for clarity as well.

8) Supplementary Information, Section IV.: Figs.S4 and S5 should be already referred to in the end of the first paragraph, as the text in the following enumeration of the three different models refers to black, blue, and red lines, respectively.

Done.

9) Supplementary Information, Equation (S39): The superfluous "d" in the integral in the denominator should be deleted.

Done.

Answers to Referee #2:

The manuscript reports on the observation of resonant Andreev tunneling processes between a Shiba state located on the superconducting tip and a superconducting substrate.

Such resonant processes have been observed previously by the same group (Huan et al., Karan et al, Kot et al) and other (Peters et al; Ruby et al.).

This resonant tunneling process can occur at an energy $\Delta + \varepsilon$, leaving the Shiba into its ground state or occurs at $\Delta - \varepsilon$, leaving the Shiba into its excited state.

Both processes have been observed previously, in particular (Peters et al; Ruby et al.). In those past experiments, the excited state could only be reached through thermal excitations.

In the present manuscript, the authors show that the excited state can be reached at very low temperature by applying a microwave signal and concluded that this observation is interesting as it paves the way toward using microwave signals to manipulate the quantum state of the Shiba.

The work is of high quality, well written and consistent with past works of the same group and others. While the manuscript presents only a small additional novelty with respect to previous works, I recommend publication but I suggest that the authors complete the manuscript along these lines :

We thank the reviewer for their positive feedback, in particular acknowledging the novelty of our demonstration that the resonant tunneling process can be accessed using microwaves instead of high temperatures.

- it would be useful to add details about the nature of the magnetic impurity in the present manuscript, to avoid going back to past published works.

We have added more details of the nature of the impurity in the supplementary information.

- with respect to quantum manipulation, the orbital and spin degeneracy of the Shiba is important. In Hay et al. works on nanowires, spin-orbit coupling is needed to lift the spin-degeneracy. What is known about the orbital and spin degeneracies of the Shiba state in the present manuscript.

We understand that this is a valid question for superconducting bound states in general. Here, we would like to point out that Yu-Shiba-Rusinov states are intrinsically spin nondegenerate. In our particular case, we have many indications that the YSR states are from a spin-1/2 impurity as discussed in Ref. 4, 7, 18, as well as arXiv:2304.02955 and arXiv:2212.11332.

- Providing a sketch of the protocol that could be employed to produce quantum manipulation of the Shiba state to observe, for example, Rabi oscillations, would reinforce the claim that quantum coherent manipulation of the Shiba state with microwave signals is possible.

The challenge here is not so much to sketch a protocol. This would probably not be much different from what is employed already in other contexts. The next step would be to expand to a system that allows for coherent coupling, e.g. between two YSR states, such as in Ref. 4. We hope that the Referee agrees that this goes beyond the scope of this paper.

minor corrections are : Figure 1 : “Tunnelling” written with three n

Done.

« proof-of-principle » i missing

Done.

Answers to Referee #3:

The authors of this collaboration report excitation of Yu-Shiba-Rusinov states in an atomic-scale tunneling junction using microwaves. Along with careful measurements of the excitation spectra, they provide a thorough theoretical analysis. This analysis goes beyond the previously used Tien-Gordon theory and accounts for most features in the spectra and, in particular, the series of (red) peaks identified in Fig. 1b. The latter are explained in terms of tunneling mediated by excited states.

Although this experiment is not the first of its kind and similar results have been reported in Ref. 9 by Peters and coworkers, the quality of the experimental data is impressive and the theoretical analysis thereof is fairly thorough (within the assumptions made by the authors, see below). Before publication may be granted, the authors should provide further explanations on a number of issues described below.

We thank the reviewer for sharing their overall positive impression. We are pleased to hear their praise regarding the quality of the data.

The main issue with this study concerns the rather “ad hoc” preparation of the “magnetic impurity” in the tip of the STM, which is subsequently studied under microwave irradiation. The preparation procedure is briefly described in the first section of the supplementary information (SI), but additional details are necessary. This group has used this method in previous work and they refer to this fact in the manuscript. However, in order to fully justify their theoretical treatment _independently of the fact that the latter agrees well with the experimental observations_ a reassessment of some of their assumptions would be most welcome. For example, what is the spin of the impurity?

We believe that the spin of the impurity is $1/2$. We have collected multiple indications over time that allow for this conclusion: the behavior of the Kondo effect in a magnetic field, tunneling between YSR states and the fact that we only see one YSR state in the gap. These findings were discussed already in a number of publications in Ref. 4, 7, 18, as well as arXiv:2304.02955 and arXiv:2212.11332. They all lead to a consistent picture for a spin- $1/2$ impurity. We have added a corresponding statement in the Supplementary Information.

In their theoretical model, the impurity is described using a minimal Anderson model in the mean-field approximation (Refs. 1 and 6 of the SI), which means the impurity is treated as a classical local exchange potential. However, in a fully quantum mechanical treatment, the impurity orbital may contain more than one electron resulting in higher spin and other terms like single-ion anisotropy. This would lead to additional excitation channels in the presence of microwaves. What is the experimental evidence that such channels are not present or relevant?

As mentioned in the previous point, we have multiple experimental indications that the spin of the impurity is $1/2$, which so far has led to an overall consistent picture. This

would exclude a scenario with multiple electrons and anisotropy or additional excitation channels. We completely agree with the referee that higher order spins, anisotropy and the like would very much complicate the interpretation, which is why we are working with this comparatively simple spin-1/2 system, which allows us to apply the simple Anderson impurity model.

More specifically, what would be the difference between a spin-1/2 impurity as assumed and a higher spin impurity with positive single-ion anisotropy for which only its lowest two spin states are available in the absence of microwaves but higher spin states being excited upon irradiation?

Exciting a higher spin state would only be possible in resonance, i.e. for a particular microwave frequency. Therefore, we could move in and out of resonance by changing the microwave frequency as we have done in Fig. 3. We would expect additional features in resonance, which we do not see in our data. The absence of such features supports our choice of model. This could be an easy way to distinguish these processes.

Can the authors entirely rule out that the internal excitations of the impurity may account for the experimental observations?

This is a generalization of the previous question. Again, we would expect such internal excitations to only happen in resonance at specific frequencies, which could be distinguished in a frequency dependent measurement, such as in Fig. 3. As we do not see any additional peaks that cannot be explained by the model, we can rule out such a scenario.

The authors also provide some evidence that supports their interpretation that processes responsible for the red peaks involve excited states. However, the discussion of why it is possible to neglect thermally activated processes does not appear entirely convincing.

It is a simple argument to exclude thermally activated processes here. If the peaks that we connect with microwave activated tunneling were thermally activated, the corresponding thermal peak at zero microwave amplitude would have to be as high as the direct tunneling peak (cf. Fig. 3d) in order to get a peak height matching with the experiment at higher microwave amplitude. This would imply a temperature much higher than the transition temperature of the superconductor, such that the system would be normal conducting. Since we observe a superconducting junction up to the highest amplitudes, we conclude that a thermal activation cannot explain the experimental data. This argument can be easily understood by looking at the corresponding Boltzmann factors.

The argument found on page 3 seems to rely on the assumption that the temperatures of irradiated and non-irradiated junctions stays roughly the same (0.56K). However, irradiation is expected to cause heating of the junction, as the authors themselves acknowledge on page 2. Therefore, they should provide an improved discussion of this point beyond the sentence (page 2) “We note that the temperature of the junction only increases by a few mK, which we can safely assume to be constant in line with previous work [8]”. Can they possibly explain, for instance, how this “few mK” estimate obtained? And why should the distribution of the irradiated atomic junction remain thermal at all?

During the measurement we can read the temperature from a thermometer at the scan-head, which increases by about 20 mK at the highest intensities. This is not enough to attribute the excited state tunneling to thermal activation. Also, as it was demonstrated in Ref. 8, the effective temperature of the superconductor does not increase as much even if the surrounding environment is heated by the microwave radiation. In addition, all theoretical calculations were done assuming a temperature of 560 mK, which fits well with the experimental data. Higher temperatures here would also lead to a substantial broadening of the peaks (cf. Ref. 8). Also, if the excited state tunneling were thermally activated, we would not observe such a stepped activation threshold as indicated in Fig. 4(a). We, therefore, believe that our interpretation is overall consistent. We have adapted the statement in the text accordingly.

Typos: in the last paragraph of page 2, the word “processes” appears twice.

Done.

REVIEWERS' COMMENTS

Reviewer #1 (Remarks to the Author):

Author: Janis Siebrecht et al.

Title: Microwave Excitation of Atomic Scale Superconducting Bound States

I read through the replies to my own comments/criticisms and the ones of the other two Referees #2 and #3 concerning the manuscript by Janis Siebrecht et al., and through the according revisions of the manuscript and supplementary. The answers to all questions and criticisms are very convincing and the changes to the manuscript and supplementary have been done to my full satisfaction.

Both other Referees also recommended publication of the manuscript after, from my point of view, minor changes. In particular, in agreement with the authors, I also don't see the necessity to know the exact nature of the magnetic impurity on the tip, as asked by Referees #2 and #3. The authors collected multiple proves for the spin 1/2 character of the impurity in a number of previous publications, which are all consistent, and exclude the additional complexity of multiple orbital physics and anisotropies. I also agree with the authors that their conclusion, that the additional peaks can only be due to the microwave irradiation, is sound. They can firmly exclude thermally activated processes. I finally agree with the authors, that a sketch of a quantum manipulation protocol asked by Referee #2 is not really essential and should be omitted.

Finally, I would like to contradict the statement of Referee #2: "While the manuscript presents only a small additional novelty with respect to previous works...". From my point of view, the successful microwave-excitation of YSR states of atomic spins coupled to superconductors, which the authors demonstrate here for the first time, is an essential step, and opens up possibilities for a vast bunch of experiments related to many ideas on the coherent manipulation of spins coupled to superconductors towards the goal of the realization of a YSR qubit.

In summary, I recommend acceptance and publication of the current version of the manuscript in *Nature Communications*.

Reviewer #2 (Remarks to the Author):

The authors provided proper answers to my questions and comments. I believe that the manuscript can be published as it is.

Reviewer #3 (Remarks to the Author):

The authors have successfully addressed my concerns and therefore, I have no objection to the publication of their manuscript.